# The Sarcoptic Mange in Maned Wolf (*Chrysocyon brachyurus*): Mapping an Emerging Disease in the Largest South American Canid

**DOI:** 10.3390/pathogens12060830

**Published:** 2023-06-15

**Authors:** Flávia Fiori, Rogério Cunha de Paula, Pedro Enrique Navas-Suárez, Ricardo Luiz Pires Boulhosa, Ricardo Augusto Dias

**Affiliations:** 1Instituto Pró-Carnívoros, Atibaia 12945-010, Brazil; flaviafiori@usp.br (F.F.); rogerio.paula@icmbio.gov.br (R.C.d.P.); r.boulhosa@gmail.com (R.L.P.B.); 2Department of Preventive Veterinary Medicine and Animal Health, School of Veterinary Medicine, University of Sao Paulo, São Paulo 05508-270, Brazil; 3Centro Nacional de Pesquisa e Conservação de Mamíferos Carnívoros-CENAP, Instituto Chico Mendes para a Conservação da Biodiversidade—ICMBio, Ministry of Environment and Climate Change, Atibaia 12952-011, Brazil; 4Department of Pathology, School of Veterinary Medicine, University of Sao Paulo, São Paulo 05508-270, Brazil; pedroenasu@gmail.com

**Keywords:** maned wolf, *Chrysocyon brachyurus*, sarcoptic mange, *Sarcoptes scabiei*, conservation, emerging disease

## Abstract

The maned wolf (*Chrysocyon brachyurus*) is the largest South American canid. In Brazil, as in other countries, it is considered an endangered species. Habitat loss, landscape changes, hunting, and roadkill are the main threats to this species. In addition, invasive diseases of domestic animals are considered to be an emerging threat to the maned wolf, where parasitic diseases are relevant. Sarcoptic mange is a skin disease caused by the mite *Sarcoptes scabiei*. This disease is currently almost globally distributed, with a remarkable host diversity. In Brazil, reports of sarcoptic mange in wildlife include several species, both wild and captive. However, the impact of this disease on wildlife is unknown. At the time of writing, there is only one published report of sarcoptic mange in maned wolves. This study sheds light on the occurrence of sarcoptic mange in free-ranging maned wolves in their natural range. A total of 52 cases (suspected and confirmed) of sarcoptic mange were identified through social media review, camera trapping, chemical immobilization and sample collection. These cases were distributed in southeastern Brazil, in the states of São Paulo (n = 34), Minas Gerais (n = 17), and Rio de Janeiro (n = 1), demonstrating a rapid and widespread spread of this disease, although it still only occurs in part of the species’ range. We expect that these results will help to subsidize future actions relevant to the control of this emerging disease.

## 1. Introduction

The maned wolf (*Chrysocyon brachyurus*) is the largest native South American canid, weighing up to 30 kg and inhabiting the central part of South America, especially the Cerrado, Chaco and Pampa biomes. Its diet consists primarily of the fruits of a typical savanna shrub, the wolf apple (*Solanum lycocarpum*), and secondarily, of rodents, birds [1], and invertebrates [2]. Despite this, individuals exhibit a dispersed social system, i.e., are essentially solitary, which is maintained by avoidance (barking), visual threat displays or fecal pheromones [2]. Adult home ranges vary across the distribution area but in natural habitats can be as large as around 80 km^2^, with little overlap with the home ranges of not paired individuals [3,4]. Mating pairs eventually locate close to each other simultaneously in the fall, but maintain a distance of >0.5 km from each other, regardless of the time of the day [3]. Publications on interspecific interactions with vertebrates other than their prey are scarce.

In Brazil, this species is classified as vulnerable (VU) [4]. Globally, however, it is classified as near threatened (NT) [5]. Habitat loss and continuous landscape changes due to agriculture and ranching, mining and other human activities, road mortality, retaliatory hunting (as a result of predation of domestic birds), and transmissible diseases are the main threats to the species [4]. Currently, transmissible diseases play an important role in the decline of carnivore populations, and are considered a major threat [6,7].

Direct contact between pets and wildlife is known to promote the bi-directional flow of pathogens [8]. The impact of domestic animal pathogens on wildlife is still unclear. However, some examples demonstrate their detrimental effects (morbidity and even mortality), such as canine distemper (CDV) outbreaks in African wild dogs in the Serengeti [9], and in the Mexican wolves in the USA [10]. There are also increasing reports of endoparasites such as *Dirofilaria imitis* [11], *Toxocara canis* [12], and *Tricuris vulpis* [13] in wild canids. Several pathogens have been reported in the maned wolf, some of which are of domestic origin [14,15,16,17,18,19,20,21,22,23,24].

Sarcoptic mange is a parasitic disease caused by the mite *Sarcoptes scabiei*. It is almost globally distributed, and has been described in humans and animals (domestic and wild) [25,26]. In Brazil, sarcoptic mange has been reported in wild [27,28,29,30] and captive animals [31]. However, despite its wide (geographic and species) distribution, reports of sarcoptic mange in Brazilian wild carnivores are rare. Sarcoptic mange has been described in humans since the mid-13th century and only later, in the 19th century, in livestock [32]. The entire life cycle of the ectoparasite occurs within the host: the females form burrows in the *stratum corneum* of the skin, then emerge to mate on the surface and lay eggs, which take 3 to 4 days to hatch [33]. The presence of females in the skin’s *stratum corneum* and their secretions cause a hypersensitivity reaction in the host, resulting in severe pruritus [34]. Depending on the development of clinical signs, the host may die from secondary infections and/or cachexia [35].

Sarcoptic mange has been described in crab-eating fox (*Cerdocyon thous*) [29,30,36], chilla fox (*Lycalopex griseus*) [37], sechuran fox (*L*. *sechurae*) [38], bush dog (*Speothos venaticus*) [39,40], and maned wolf (*C*. *brachyurus*) in Bolivia [41] and Brazil [39]. However, all reports are case descriptions. Prior to the present work, there is only one published report of sarcoptic mange in the maned wolf [41]. Nevertheless, the aim of this study was to collect information on the occurrence of sarcoptic mange in free-ranging maned wolves in their natural range in Brazil.

## 2. Methodology

### 2.1. Study Design, Study Area and Study Period

This study included confirmed cases (group G1, hereafter G1) and suspected cases (group G2, hereafter G2). G1 individuals were captured with box traps or rescued from September 2017 to February 2023. In addition, we included in this group a record traced back to January 2008 (the first record of sarcoptic mange in maned wolves, to our knowledge [41]). For the G2, camera trap images were obtained (G2A) and an active social media search was conducted (G2B). The increasing number of confirmed cases (G1) motivated the expansion of the search for new potential cases (G2). G2A individuals were enrolled from December 2016 to February 2023, and G2B individuals were enrolled from October 2017 to March 2023. G1 and G2A animals were monitored by the Pró-Carnívoros Institute and the National Research Center for Carnivore Conservation (CENAP) of the Chico Mendes Institute for the Biodiversity Conservation (Brazilian Ministry of Environment and Climate Change). All the data collected were in accordance with the actions of the National Action Plan for Wild Canids (https://www.gov.br/icmbio/pt-br/assuntos/biodiversidade/pan/pan-canideos (accessed on 16 of January 2023)). The study area for G1 and G2A included the state of São Paulo, along the border with the state of Minas Gerais, in southeastern Brazil (Figure 1). For G1, 14 sampling points were monitored and for G2A, 49, according to the activities of the “Lobos do Pardo” project and indications of suitable habitats for the presence of maned wolves. Records for G2B were obtained elsewhere in the maned wolf range, in Brazil. The maned wolf’s range was obtained in the IUCN Spatial Data Download (https://www.iucnredlist.org/resources/spatial-data-download (accessed on 17 of January 2023)) and all cases were plotted on a thematic map prepared in the QGIS version 3.20 (Odense) software.

### 2.2. Confirmed Cases (Group G1)

Since there are no pathognomonic lesions of sarcoptic mange, a case was considered confirmed if mites were identified in skin scrapings or biopsies. One case included in this study was previously described by Jorge and Jorge [39].

The number of box traps used simultaneously in each sampling site varied from 2 to 6 (mean = 4). The box traps were used for physical containment and were distributed over the monitored sampling site at pre-determined locations. A dose of 4 mg/kg of Zoletil 50^®^ (tiletamine hydrochloride 125.0 mg and zolazepam hydrochloride 125.0 mg) was used for chemical containment. The anesthetic was administered intramuscularly, using a hypodermic syringe. Clinical parameters (heart rate, respiratory rate, rectal temperature, saturation, and eyelid, anal, and digital reflexes) were monitored throughout the procedure. A general condition assessment was performed after chemical containment, and age was estimated based on the dental arches [42,43,44]. Weight was assessed and compared to a scale proposed by Emmons [45] and ICMBio [46]. Animals were fully examined for skin lesions and deep skin scrapings, and punch biopsies were collected under chemical restraint. The scrapings were placed on slides and later examined under a light microscope to look for mites. For histopathologic examination, the skin fragments were fixed in 10% neutral buffered formalin, embedded in paraffin wax (FFPE), sectioned at 5 μm, and stained with hematoxylin and eosin for light microscopy. Morphological identification of Acari in tissue sections was based on published guidelines [47]. Captured wolves were monitored with radio collars throughout the study period. After anesthetic recovery, the animals were released at the same capture site.

### 2.3. Suspected Cases (Group G2)

For this assessment, we use camera trap surveys (G2A) and social media posts (G2B) containing descriptions or photographs of maned wolves with suggestive mange skin lesions. For G2A, a total of 71 different sites were used for camera trapping during the study, with the number of simultaneous sites varying depending on the season and the duration of each field campaign. A number of 30–35 cameras were left in fixed positions throughout the entire sampling period and 15–20 cameras were repositioned after one year. In the daytime images captured by camera traps, alopecia could be identified from the frames of the recorded videos. For the night images, alopecia was observed as peculiar dark patches on the exposed skin in contrast to the animal’s fur.

For G2B, we considered records whose published images contained one or more clinical signs of the disease, such as alopecia, scabbing, desquamation, and skin lesions (due to pruritus) at the elbows, hocks, and/or margin of the pinna. The authors of the social media posts (from governmental institutions, NGOs, veterinarians or biologists) were contacted and asked if, in addition to the suggestive clinical signs, pruritus was observed, as well as the exact location of the recording.

In addition, if identification of the mite in skin scrapings or biopsies from the captured wolves (G1) was not possible, and only the presence of the suggestive lesions was observed, these animals were included in G2A.

## 3. Results

A total of 52 cases of sarcoptic mange were identified, of which 10 (19.2%) were confirmed (G1) and 42 (80.8%) were suspected (Table 1, Figure 1), including 12 (28.6%) from G2A and 30 (71.4%) from G2B. According to the origin, 34 (65.4%) cases were from the state of São Paulo (including all the animals from G1, in addition to 11 of animals from G2A and 13 from G2B), 17 cases from the state of Minas Gerais (one from G2A and 16 from G2B) and one case from the state of Rio de Janeiro (G2B). Figure 2 shows examples of day and night images with the characteristic signs of sarcoptic mange.

Of the confirmed animals (G1), eight (53%) were males and seven (47%) were females. The body condition of 13 (86.7%) animals was within the normal body weight range for the species, including 3 juveniles (Appendix A). In six cases, mites were seen in the biopsies, and in four animals, the deep skin scraping was positive. Of the 42 animals in G2, 7 (16.7%) were captured with camera traps in the study area, and in 5, no mites were observed in skin scrapings and biopsies (11.9%) (G2A), and 30 (71.4%) were collected indirect data collection (G2B).

According to the period in which records were noticed, considering G1 and G2: in 2008, one case; from 2009 to 2014, no record was made; in 2015, one case; in 2016, one case; in 2017, five cases; and in 2018, one case. The records became abundant from 2019 on, with six cases; in 2020, nine cases; in 2021, eight cases; in 2022, sixteen cases; and in 2023, four cases were recorded by March.

The trapping effort (G1) was 18,312 traps × hours. A total of 67 captures of 20 individual maned wolves occurred at all 14 monitoring sites. The sampling effort for the camera trap survey (G2A) was 870,168 traps × hours, resulting in 731 images of 43 individual maned wolves (adult and juveniles), at 27 sampling sites (55% of the monitored sampling sites).

Microscopically, in three cases, the epidermis showed marked parakeratotic hyperkeratosis, degenerate and non-degenerate neutrophils, and multifocal areas of ulceration associated with numerous intralesional adult mites. Within the dermis, a mild mixed inflammatory infiltrate (eosinophils, neutrophils, plasma cells with fewer macrophages and lymphocytes), congestion of deep dermal vessels, and lymphangiectasia were observed. Morphologically, the mites measure between 200 × 400 μm, have articulated appendages, a chitinous exoskeleton, dorsal spines, striated muscle, intestinal and reproductive structures, and a body cavity (Figure 2).

## 4. Discussion

The present study described the occurrence of sarcoptic mange in maned wolves in southeastern Brazil. The methodology was limited by the small number of skin scrapings and biopsies obtained from affected animals, which prevented a comprehensive understanding of the dermatological manifestations and maintenance of the disease in this species. Despite these limitations, it is important to emphasize the commendable sampling effort undertaken in this study, which allowed the collection of valuable data on sarcoptic mange in maned wolves.

The increasing number of individuals affected by sarcoptic mange reported in a small portion of the maned wolf’s range in Brazil (approximately 6% of the total species’ total range in the country), suggests that this disease is a new emerging threat to the species, especially in anthropized areas. The first report of sarcoptic mange in a maned wolf in Brazil was recorded in 2008 [39]. In this study, all possible sources of information were searched for suspected or confirmed cases of sarcoptic mange in maned wolves throughout the species range. From 2008 to 2019, only a few records of animals with clinical signs of sarcoptic mange were registered. The vast majority of cases were recorded in the study area after 2019. With the exception of the records from the states of Minas Gerais (Cbr 33, 38) and Rio de Janeiro (Cbr 42), the recorded cases from the state of São Paulo represent the core zone of the disease circulation. In contrast to maned wolves, records of sarcoptic mange in other wild canids, such as crab-eating foxes and bush dogs, are distributed throughout their range [29,30,36,39,40].

Observations in G1 animals revealed areas of alopecia and crusts on the skin as clinical signs. However, skin samples from five animals with clinical signs were negative in the parasitological and histopathological diagnoses, and were included in the G2A group. Although a proper protocol for mite isolation has not been established, it has been suggested that the collection of mites depends on the stage of infestation, as they may be difficult to collect at certain stages, leading to false negative results in the laboratory techniques used [35,48]. Thus, our results suggest that animals in the early and late stages of infestation are diagnosed by skin scraping. On the other hand, animals in the intermediate stage of the infestation are diagnosed by skin biopsy for histopathologic analysis.

Individuals Cbr 5, Cbr 8 and Cbr 9 (which were negative in skin scraping) all had a history of direct (pairs during breeding season or parents with youngsters) and/or indirect contact (high home range overlap) with other wolves positive for sarcoptic mange (Cbr 6, 7, 14 and Cbr 10, 11, 12, respectively). This contact was recorded using GPS collar monitoring installed on the animals. Once it was known that the mange transmission occurs either directly, through physical contact, or indirectly, through fomites [49,50], clinical diagnosis was applied to these animals. Both wolves (Cbr 5 and Cbr 9) showed characteristic clinical signs of sarcoptic mange such as alopecia, scaling, and crusting of the skin, in addition to clinical cure after in loco treatment with Bravecto^®^ (25–65 mg of fluralaner/kg of body weight). Although these animals showed improvement in the clinical signs, the diagnosis of sarcoptic mange could not be confirmed because Bravecto^®^ (Merck & Co., Inc., Rahway, NJ, USA) does not specifically target *Sarcoptes scabiei*.

Animals included in the G2 group were classified as suspect because the images showed clinical signs suggestive of sarcoptic mange [26,35]. Signs such as alopecia, crusts, and scaling (the latter two clinical signs were only visible in the daytime images) were observed in camera trap images. As the camera traps were used for a long-term survey in the study sites, the individuals included in the study could be registered over several months, allowing the analysis of the evolution of the infestation process. Thus, the observations from different stages of infestation of the animals in groups G1 and G2A served as a basis for comparing the clinical status of the wolves included in G2B. G2B animals were separated spatially and temporally, reducing the likelihood of duplicate cases. In addition, the maned wolf is not a gregarious species and therefore does not form permanent groups. For this reason, relationships between individuals living in the same groups or not and passing the mite among themselves are unlikely.

All cases of sarcoptic mange (G1 or G2) were recorded in areas where the landscape is changing due various human activities, especially agriculture and cattle ranching [51]. In addition, camera trap data show an overlap in the land use between maned wolves (groups G1 and G2A) and domestic animals such as dogs and cattle, two species commonly affected by sarcoptic mange [52]. Indirect contact between wild and domestic canids in all study areas may suggest the potential transmission of the mites through the modified areas. However, at the other study sites in other parts of Brazil, camera trapping also revealed that maned wolves and domestic animals shared anthropized areas, and yet there are no records of infested wild canids, except for crab eating foxes and sometimes bush dogs. In addition, according to Sinclair et al. [53], environment change due to climate change, the movement and migration of hosts, and the presence of new pathogens may bring both domestic and wild species into closer contact in areas where this would not have been possible. This process favors the emergence of infectious diseases [53].

The possibility of anthropogenic changes influencing the epidemiology of sarcoptic mange is suggested. As with domestic dogs and cattle, there are other species that could be part of the epidemiological cycle of mange. Exotic species may be affected and transmit the parasites. Examples include the wild boar (*Sus scrofa*), which has been reported to have mange in its natural range [54,55], and lagomorphs such as the European hare (*Lepus europaeus*), which, although not reported to have mange, has mange in other members of the genus [56]. These exotic species are found in different regions of Brazil, in sympatry with the maned wolf [57,58]. Therefore, epidemiologic surveillance is proposed for these four domestic and exotic species, considering their contribution to the epidemiology of sarcoptic mange.

## 5. Concluding Remarks

The records compiled in this study and the mapping of maned wolf sarcoptic mange are a first step in further evaluating how an emerging disease may expand its range in relation to variables related to human’s presence. The information generated leads to a negative alert regarding a high impact on the health of an endangered species. This emerging disease must be a high priority level for maned wolf conservation strategies. Once it has been established that the epidemiology of sarcoptic mange can be linked to landscape and bioclimatic variables, as well as to close contact with invasive and domestic species, preventive strategies must be put into practice; nevertheless, genetic studies of mite populations must be carried out to elucidate the transmission chain of sarcoptic mange involving maned wolves and other hosts, whether domestic (e.g., dogs) or wild, whether invasive or native. In particular, educational efforts focused on proper land use and responsible care of the domestic animals are important issues to be developed for the conservation of the maned wolf. Disease surveillance of the Brazilian fauna is gradually being standardized and professionalized. In the case of the native wild canids, these actions are being discussed in the National Action Plan for Wild Canids of the Instituto Chico Mendes de Conservação da Biodiversidade (ICMBio) of the Brazilian Ministry of Environment and Climate Change, and the results will be useful in prioritizing public policies to help reduce the spread of sarcoptic mange in maned wolves.

## Figures and Tables

**Figure 1 pathogens-12-00830-f001:**
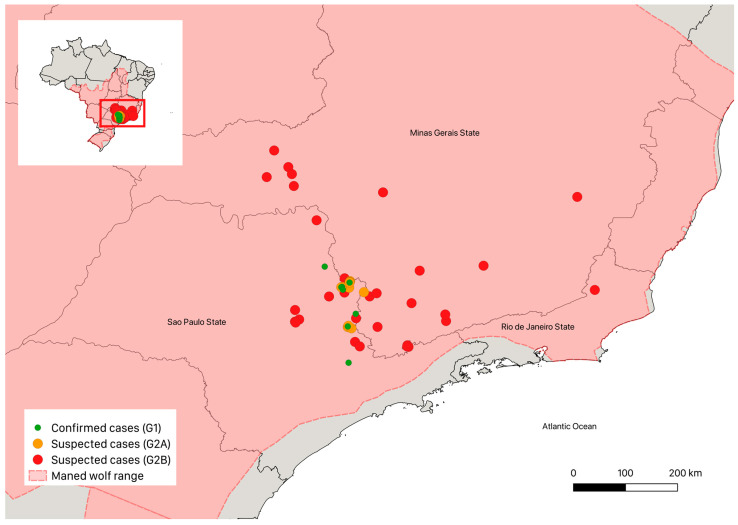
Spatial distribution of confirmed and suspected cases of sarcoptic mange in maned wolves in Brazil.

**Figure 2 pathogens-12-00830-f002:**
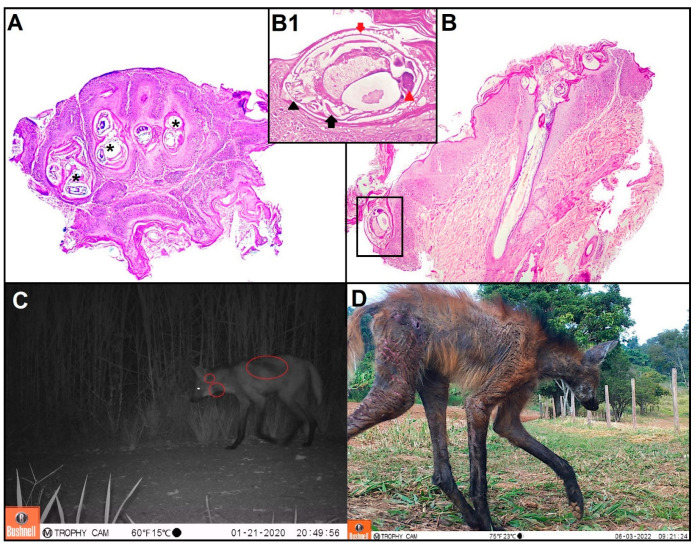
(**A**). BTH 986. Skin. Numerous intralesional adult mites (asterisk). HE, 40× (**B**). BTH 986. Skin. Numerous intralesional adult mites. HE, 40× (**B1**) (Inset). Note a female specimen of *Sarcoptes scabiei* in the superficial epidermis. The mites have jointed appendages (black arrowhead), a chitinous exoskeleton (red arrow), ovary (red arrowhead), and striated muscle (black arrow). HE, 400×. (**C**). Nighttime image showing a maned wolf with dark spots in areas of the coat, indicating alopecia. (**D**). Daytime image showing a maned wolf with areas of alopecia, crusts and cracks in the skin.

**Table 1 pathogens-12-00830-t001:** Sarcoptic mange cases in maned wolves, according to identification groups.

Identification Groups		Individuals	Date of Record	Municipality	State
			(mm/yyyy)		
G1		Cbr 1	05/2022	Mococa	São Paulo
		Cbr 4	01/2008	Itatiba	São Paulo
		Cbr 6	07/2019	Mococa	São Paulo
		Cbr 7	04/2021	Mococa	São Paulo
		Cbr 10	12/2017	Mogi-Guaçu	São Paulo
		Cbr 11	12/2017	Mogi-Guaçu	São Paulo
		Cbr 12	12/2017	Mogi-Guaçu	São Paulo
		Cbr 13	04/2021	São João da Boa Vista	São Paulo
		Cbr 14	09/2018	São José do Rio Pardo	São Paulo
		Cbr 48	02/2023	Cajuru	São Paulo
G2	G2A	Cbr 2	06/2022	Mococa	São Paulo
		Cbr 3	08/2022	Mococa	São Paulo
		Cbr 5	03/2019	Mococa	São Paulo
		Cbr 8	04/2021	Mococa	São Paulo
		Cbr 9	09/2017	Mogi-Guaçu	São Paulo
		Cbr 15	05/2022	Mococa	São Paulo
		Cbr 16	09/2021	Mococa	São Paulo
		Cbr 17	06/2020	Itapira	São Paulo
		Cbr 18	04/2020	Mogi-Guaçu	São Paulo
		Cbr 19	11/2020	Mogi-Guaçu	São Paulo
		Cbr 20	10/2019	São José do Rio Pardo	São Paulo
		Cbr 21	03/2020	Botelhos	Minas Gerais
	G2B	Cbr 22	10/2022	Itirapina	São Paulo
		Cbr 23	07/2020	Santo Antônio do Jardim	São Paulo
		Cbr 24	04/2022	São Bento do Sapucaí	São Paulo
		Cbr 25	05/2022	São Bento do Sapucaí	São Paulo
		Cbr 26	11/2021	São Bento do Sapucaí	São Paulo
		Cbr 27	12/2020	São Carlos	São Paulo
		Cbr 28	03/2022	São José do Rio Pardo	São Paulo
		Cbr 29	03/2019	Serra Negra	São Paulo
		Cbr 30	02/2021	Arceburgo	Minas Gerais
		Cbr 31	07/2022	Botelhos	Minas Gerais
		Cbr 32	04/2022	Campos Altos	Minas Gerais
		Cbr 33	03/2020	Córrego Novo	Minas Gerais
		Cbr 34	11/2022	Uberlândia	Minas Gerais
		Cbr 35	05/2016	Jardinésia	Minas Gerais
		Cbr 36	09/2020	Pouso Alto	Minas Gerais
		Cbr 37	03/2017	Santana da Vargem	Minas Gerais
		Cbr 38	01/2021	São João del Rei	Minas Gerais
		Cbr 39	01/2019	São Gonçalo do Sapucaí	Minas Gerais
		Cbr 40	08/2022	Soledade de Minas	Minas Gerais
		Cbr 41	07/2015	Uberlândia	Minas Gerais
		Cbr 42	07/2022	Santo Antônio de Pádua	Rio de Janeiro
		Cbr 43	07/2022	Uberaba	Minas Gerais
		Cbr 44	07/2021	Uberaba	Minas Gerais
		Cbr 45	12/2022	Inconfidentes	Minas Gerais
		Cbr 46	01/2023	Itirapina	São Paulo
		Cbr 47	10/2020	Monte Alegre do Sul	São Paulo
		Cbr 49	12/2022	Itirapina	São Paulo
		Cbr 50	01/2019	Pedregulho	São Paulo
		Cbr 51	03/2023	Campestre	Minas Gerais
		Cbr 52	01/2023	Tambaú	São Paulo

## Data Availability

Data available on request due to restrictions.

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
