# Peer review of "The Sarcoptic Mange in Maned Wolf (Chrysocyon brachyurus): Mapping an Emerging Disease in the Largest South American Canid"

_pathogens, 2023, doi:10.3390/pathogens12060830_

Round 1
Reviewer 1 Report
Dear authors,
thank you for submitting your manuscript to Pathogens journal.
I have the following comments-suggestions:
Introduction.
Line 57. Please, change the word "tunnels" to "burrows" and the words "remain with" to "laid".
Line 59. Please, change the word "itching" to "pruritus".
Methodology.
Confirmed cases section.
Since there are no pathognomonic lesions of sarcoptic mange, the presence of macroscopic lesions cannot be included as a confirmatory methodology The 5 cases where there was a "clinical diagnosis" should be included in G2.
Suspected cases section.
Again, since there are no pathognomonic lesions of sarcoptic mange, the presence of alopecia, scabs, scaling, and skin wounds cannot be considered to be suggestive of sarcoptic mange, especially, when the most prevalent clinical sign of sarcoptic mange is pruritus. Even in the presence of severe pruritus, the differential diagnosis would be wide. The presence of pruritus and a "more suggestive skin lesions distribution" (elbows, hocks, margin of the pinna) should be considered as criteria. In addition, in this section, you should explain what considerations/criteria were taken to include wolves in the G2B group.
Results section.
Based on the above, the results must be changed.
Legend Figure 2A. From the low magnification picture you have included it is impossible to observe parakeratosis. Explaining the presence of the mites is just enough.
Discussion section.
Discuss in more detail the limitations of the study: a low number of scrapings, dermatological evaluation, and biopsies. The methodology is not accurate and cannot support the results.
According to your methodology, was there any possibility of duplicate cases? Were all wolves included in this study identified as single individuals? By what methods? Please, clarify this.
You barely mention antiparasitic treatment (line 217). How many animals received this treatment? Response to treatment can be considered as a confirmatory criterion for many of the cases included, but you did not mention treatment as part of your methodology. This is of paramount importance.
None
Author Response
Dear reviewer, thank you for your comments. Please find below, the point-by-point response to each comment.
Introduction:
Line 57: Suggestions accepted. Rewritten.
Line 59: Suggestion accepted. Rewritten.
Methodology:
Confirmed cases section:
We agree that the 5 cases where there was a “clinical diagnosis” could not be included in G1. The inclusion criteria only included the identification of the mite in skin scrapings or biopsies. Rewritten.
Suspected cases section:
We agree with your comments. This section was completely rewritten. Despite the camera trap survey performed by our team, the authors of posts in social medias that contained images of manned wolves with suspected lesions were contacted, and provided more details about the affected animal, including the presence of pruritus and the location of the record.
Moreover, when the identification of the mite in skin scrapings or biopsies from the captured wolves (G1) was not possible, and only the presence of the suggestive lesions was observed, these animals were included in G2A.
Results:
Results were rewritten based on the changes in methods.
Figure 1 was reformulated, singe 5 individuals from G1 were moved to G2A.
Legend of figure 2A changed according to suggestion.
Discussion:
A paragraph was included highlighting the limitations of the study.
A comment about the low probability of duplicating G2B cases was included. Moreover, G1 and (some) G2A cases were monitored by radio-collars during the study period (a comment about radio-collar monitoring was included in the Methods).
The treatment was included, but not mentioned in Methods nor used as confirmatory diagnosis, since Bravecto does not act specifically against Sarcoptes scabiei.
Thank you again for your valuable comments.

Reviewer 2 Report
This manuscript gives an important update on the epidemiological situation of a neglected emerging parasitic disease in a vulnerable wild species: the maned wolf.
The manuscript describes how sarcoptic mange can severely affect this wild carnivore and this gives merit to the authors, as studying protected wild species requires great efforts and coordination.
Overall, the English should be carefully checked and some parts of the text can be modified to improve the manuscript.
Please find below some comments and suggestions on the specific section, and line-by-line.
INTRODUCTION:
You might extend introduction section with some more information on the biology/ecology of maned wolf (eg. The diet, type of interactions with other species, mating season etc)
Line 35: Especially
Line 65-66: there is only one published report of sarcoptic mange in maned wolf.
Line 67: please change this sentence. A suggestion might be “this study aims at gathering information on the occurrence of sarcoptic mange…”
METHODS
Line 72: why G1 and G2? Isn’t it better to call them Group “confirmed” and group “suspected”?
Line 76: G2A Same problem as above, the abbreviations/letters are confusing. I suggest to simplify and consider adding a graphical legenda/table.
Line 100: please check reference “Jorge and Jorge”…is it correct?
Lines 114-115: it’s almost impossible to distinguish mite species based on hematoxylin/eosin stain after paraffin fixation, as major the body features and details (e.g. hair, genital pores etc) are lost. Morphological identification should be done on fresh or ethanol-preserved samples, directly on mite or skin scraping samples. Please explain better the methodology here and use the reference accordingly.
Table 1 can be put in Additional materials.
Same for Table 2: you can put this table in the Additional materials, leaving in the main text a table with resumed info (eg it is not necessary to list all the animals with individual information) on positive/negative wolves, sex, age, mange diagnosis type. I suggest putting in this table positive AND negative divided per diagnosis type (G1, G2A,B)
Figure 1: there is no figure caption
Figure 2: the same figure is repeated 2 times
Line 153: youngster?
DISCUSSION
In general, I suggest adding some more considerations on the possible eco-epidemiological relations between possible wild and domestic species carrying sarcoptic mange. Some questions that you could answer to implement discussion section are:
- what are the genetic strains of sarcoptic mange affecting wild canids in brazil (if there is any information?)
- what are the possible wild and domestic sources of sarcoptic mange for maned wolf?
- is it possible that some particular social or predatory behaviour can influence the transmission of sarcoptic mange in maned wolf?
- Did you record any case of zoonotic scabies in vets or rangers handling affected wolves?
- How would you improve or implement passive and active surveillance in this species?
- How would you improve the treatment protocol (if any) to treat sarcoptic mange in maned wolf?
Line 184: The increasing number of individuals affected by sarcoptic mange reported in a small portion of the maned wolf….
Line 186: periurban/urban areas
Line 187: the first report of sarcoptic mange in maned wolves in Brazil was recorded in 2008
Line 190: clinical signs
Line 191: eastern most records?
196-201 you can eliminate this part, as it is not necessary to discuss each individual sampled wolf.
Line 206-208: what do you mean by “intermediate stage” here? And why histopathological analysis should be more specific in this case?
In my experience with European wolves and wildlife species, it is very difficult to retrieve Sarcoptes mites from wolves and dogs, while foxes usually present a big amount of crusts which are plenty of mites.
Even in wolves with severe clinical signs such as extended areas of alopecia and hyperpigmentation, several skin scrapings were negative or with 1-2 mites in the whole body. I don’t know if this is the same case of maned wolf in Brazil, and this is certainly a fascinating topic to study in the future.
Line 210: not clear to me if you are referring to your sample or to general scientific literature here.
Line 228-9: brackets are not closed
Line 232: IN Brazil
Line 233: what do you mean by “disturbed areas”? maybe urban areas?
Line 234-238: the meaning of this sentence is not completely clear to me
I suggest adding a reference here, as example of interspecific transmission between wild boar and another wild ungulate species: First report of interspecific transmission of sarcoptic mange from Iberian ibex to wild boar. https://doi.org/10.1016/0304-4017(89)90064-2.
Another interesting example of circulation of sarcoptic mange in wild carnivores is in this recent paper: doi: 10.1051/parasite/2023012, in which the authors described how wild felids can share similar genetic strains of sarcoptic mange with other wild canids such as wolves and red foxes.
Line 244: please rephrase this sentence as it is not clear.
As I suggested above, the English language should be carefully checked by the authors as some sentences are not clear, and words misspelled.
Author Response
Dear reviewer, thank you for your comments. Please find below, the point-by-point response to each comment.
Introduction:
The introduction was extended, including information oh the biology of the maned wolf (diet, intra and interspecific interactions and mating season.
Line 35: Rewritten.
Line 67: Rewritten.
Methods:
Lines 72 and 76: We thought about it when writing the manuscript, but suspected cases forced us to use abbreviations, which were maintained.
Line 100: Yes, the reference is correct.
Lines 144-115: Samples were send to the pathology laboratory and processed according the reference.
Results:
We kept Table 1 and put Table 2 in Additional material. We included a note indicating the groups in which the animals were included.
Figure 1: There was a figure caption.
Figure 2: The figure was not repeated twice.
Discussion:
- No information about genetic strains of S. scabiei is available for wolves and other canids. Actually, this is a future project.
- Social or predatory behavior are probably not associated with infraspecific transmission, since the wolves are lonely most of the time.
- No cases of zoonotic scabies were recorded.
- That is a question that is being discussed in the National Plan for Conservation of Wild Canids of the Brazilian Ministry of Environment and Climate Change.
- Bravecto® (25-65 mg of fluralaner/kg of body weight) seems to be a promising treatment, but further analysis should be made.
Line 184: Rewritten.
Line 186: Maned wolves are not seen in urban areas, we changed “disturbed" to “anthropized”.
Line 187: Rewritten.
Line 190: Rewritten.
Line 191: Excluded.
Lines 196-201: Excluded.
Line 206-208: Rewritten. The difference of retrieving mites between European wolves and maned wolves is certainly an interesting topic.
Line 210: This phrase was excluded, and the treatment was discussed later in this paragraph. Rewritten.
Lines 228-9: Bracket was closed. Thank you.
Line 232: Rewritten.
Line 233: We meant anthropized (by agriculture, urban growth, etc.). Rewritten.
Lines 234-238: Rewritten.
The second reference was included. The first seemed out of context, although it is interesting.
Line 244: Rewritten.
Thank you again for your valuable comments.

Reviewer 3 Report
Authors here describe the prevalence of sarcoptic mange in maned wolves across Brazil. As the authors point out, there are hardly any studies published or carried out that have assessed the impact of this disease on wildlife. This study brings to light the occurrence of sarcoptic mange in maned wolves.
Comments:
The author talks about Farms in the manuscript. It would be of interest to readers if the author could provide a definition of farms at the beginning of the manuscript. I don't think I found the description anywhere.
It would also help readers if the author could describe how the data mining was carried out and which databases were used.
Minor corrections:
Might be an editorial issue, but I am not sure if the figure below Table 1 is a repeat. Similarly, figures on Pages 7 and 9 are repeats.
Line 62: Remove 'in South America' as on Line 64 it reads Bolivia and Brazil
Author Response
Dear reviewer, thank you for your comments. Please find below, the point-by-point response to each comment.
We called sampling points as farms, but we reverted to sampling point for clarity.
Data mining procedure was completely rewritten.
Maybe there figure repetitions were a matter of conversion from Apple Pages to Microsoft Word, but we have uploaded a single figure in the journal’s system.
Line 62: Rewritten.
Thank you again for your valuable comments.

Reviewer 4 Report
Dear Authors,
I have appreciated the article “The sarcoptic mange in maned wolf (Chrysocyon brachyurus): mapping an emerging disease in the largest South American canid”
Regarding the realization of this study I don't have major comments.
There are only a few suggestions
Best regards
Line number:
50. Sarcoptes scabiei….varietas?? expand on this point
73. change with “G1 individuals were captured with box traps”
74. Insert the term “farm” in the study area and specify what it means
87. Figure 1 should represent study area only. Figure with case confirmed and suspected should be placed in the "results" paragraph.
92. Insert name of version Qgis;
94. The caption must be placed under the figure.
112. Add “for histopathological examination”, the skin fragments….
126. double space
141. I suggest dividing figure 2 into two figures, one with histopathological exam results and one with visual exam results with camera traps
144. Table 1 - better highlight the subdivision into identification groups - try to place it on a single page or insert it as supplementary materials
147. Insert caption Figure
148. Insert caption Figure
158- 162. Better a graph
164. Double space
175. Caption figure 2, B1 is missing
193. ….the recorded case of Sao Paulo state…
230. Missing brackets
244. It is important to include some more information on the Sarcoptes scabiei varietas when discussing the infectivity of Sarcoptes among different species of animals
...see Microsatellites as markers for comparison among different populations of Sarcoptes scabiei – Soglia et al., 2010 - https: // doi: 10.4081/ijas.2007.1s.214
259. Double space
Minor editing of english language required
Author Response
Dear reviewer, thank you for your comments. Please find below, the point-by-point response to each comment.
Line 50: We did not mentioned varietas in the manuscript since the interspecific transmission was not addressed in the manuscript.
Line 73: Rewritten.
Line 74: The term farm was changed to sampling points, as other reviewer had suggested.
Line 87: It is difficult to delimit the study area, as it was a border zone between the states of São Paulo and Minas Gerais, therefore figure 1 was kept as it is.
Line 92: Name of QGIS version included.
Line 94: The Pathogens Journal does not have a specific placement for captions, but we have changed according to your suggestion even so.
Line 112: Added. Rewritten.
Line 126: Double space removed.
Line 141: Figure 2 was maintained as is.
Line 144: Table 1 was maintained in the text, and table 2 was moved to supplementary material.
Line 147-8: Inserted.
Lines 158-62: We kept the text, since the journal has changed their policies and are requiring at least 4000 words for the manuscript be accepted as full article.
Line 164: Double space removed.
Line 175: Caption inserted (it was the inset).
Line 193: Added, thank you.
Line 230: Bracket was closed, thank you.
Line 244: We did not mentioned varietas in the manuscript since the interspecific transmission was not addressed in the manuscript.
Line 259: Double space removed.
Thank you again for your valuable comments.

Round 2
Reviewer 1 Report
Dear authors, thank you for revising the manuscript. I have no comments or any other suggestions. Thank you again.
Author Response
Dear reviewer, thank you very much for your valuable comments. I'm sure our manuscript has greatly improved.
Reviewer 2 Report
Thank you for addressing my comments and suggestions.
Please check grammar and English, as there are some mistakes.
line 66: the entire life cycle
line 67: use present tense: lay, not laid.
The entire evolutionary cycle of the ectoparasite occurs within the host: the females form burrows on the stratum corneum of the skin, then emerge to mate on the surface and lay eggs, which takes 3 to 4 days to hatch.
Line 259: This process?
Please check carefully whether references and related numbers are complete and correct.
English Language should be checked by a native speaker.
Author Response
Dear reviewer, thank you for your comments. Please find below, the point-by-point response to each comment.
Grammar was throughly checked.
Lines 66-68: Thank you for the suggestion. Rewritten.
Line 259: not landscape, but anthropic changes. Rewritten.
Thank you again for your valuable comments. We are sure that your contribution has greatly improved our manuscript.